# The Cost of Alternative Water Supply and Efficiency Options under Uncertainty: An Application of Modern Portfolio Theory and Chebyshev's Inequality

Dat Tran [1,*], Tatiana Borisova [2,†] and Kate Beggs [1]

1 Florida Legislative Office of Economic and Demographic Research, Tallahassee, FL 32399, USA
2 Food and Resource Economics Department, The University of Florida, Gainesville, FL 32611, USA
* Correspondence: tran.dat@leg.state.fl.us
† Current address: Economic Research Service, U.S. Department of Agriculture, Washington, DC 20250, USA.

**Abstract:** Sea-level rise, population growth, and changing land-use patterns will further constrain Florida's already scarce groundwater and surface water supplies in the coming decades. Significant investments in water supply and water demand management are needed to ensure sufficient water availability for human and natural systems. Section 403.928 (1) (b) of the Florida Statutes requires estimating the expenditures needed to meet the future water demand and avoid the adverse effects of competition for water supplies to 2040. This study considers the 2020–2040 planning period and projects (1) future water demand and supplies; and (2) the total expenditures (capital costs) necessary to meet the future water demand in Florida, USA. The uniqueness of this study compared with the previous studies is the introduction of a probabilistic-based approach to quantify the uncertainty of the investment costs to meet future water demand. We compile data from the U.S. Geological Survey, Florida's Department of Agriculture & Consumer Services, Florida's Water Management Districts, and the Florida Department of Environmental Protection to project the future water demand and supplies, and the expenditures needed to meet the demand considering uncertainty in the costs of alternative water supply options. The results show that the total annual water demand is projected to increase by 1405 million cubic meters (+15.9%) by 2040, driven primarily by urbanization. Using the median capital costs of alternative water supply projects, cumulative expenditures for the additional water supplies are estimated between USD 1.11–1.87 billion. However, when uncertainty in the project costs is accounted for, the projected expenditure range shifts to USD 1.65 and USD 3.21 billion. In addition, we illustrate how using Modern Portfolio Theory (MPT) can increase the efficacy of investment planning to develop alternative water supply options. The results indicate that using MPT in selecting the share of each project type in developing water supply options can reduce the standard deviation of capital costs per one unit of capacity by 74% compared to the equal share allocation. This study highlights the need for developing more flexible funding strategies on local, regional, and state levels to finance additional water supply infrastructure, and more cost-effective combinations of demand management strategies and alternative water supply options to meet the water needed for the state in the future.

**Keywords:** alternative water supply; water conservation; water supply portfolio; forecasting expenditures and funding; uncertainty

## 1. Introduction

Water security, defined as sufficient access to affordable clean water for human needs and the natural environment [1], has been threatened by population growth, extreme climate events, and water pollution in many regions of the world, including the United States (U.S.). Florida—the third most populous state in the U.S.—can serve as an example of a region requiring significant investments to improve water security. The latest risk-assessment report commissioned by Florida's House of Representatives identifies water

as one of the most pressing problems facing Floridians [2]. In the next 20 years, Florida's population is projected to reach 26.4 million from about 22.2 million people today [3]. Florida depends on its beaches, freshwater springs, reefs, and national parks to help draw tourists to the state. In 2019, the state welcomed more than 131 million people—a number expected to continue an upward trend [4,5]. As a result of population and tourism growth, water demand is expected to increase considerably in the next two decades, requiring significant investments in new water supply and demand management strategies aimed at both meeting the growing water demand and protecting water resources [6]. The challenges facing future water resources management in Florida reflect similar challenges facing in economically and ecologically important coastal areas around the world under increasing uncertainties (e.g., climate extremes and dynamic socio-economic conditions).

Groundwater has provided for most of Florida's freshwater needs; however, sustainable water supply will require investing in a mix of water supply options. In 2015, Florida's public supply (PS) water utilities used 3059.83 million cubic meters (MCM) (or 38.71 percent of Florida's total 7904.31 MCM freshwater withdrawals [7,8]), and PS relied almost exclusively on groundwater. PS is Florida's largest groundwater use sector, withdrawing 2637.16 MCM of groundwater in 2015, followed by the agricultural sector with 1396.42 MCM [8]. Historically, Florida's average annual precipitation of 1365 mm [9] provided for sufficient aquifer recharge, making groundwater sources abundant and accessible. The costs of accessing groundwater for residential supply were relatively affordable. However, population growth and changing land-use patterns shifted the balance between groundwater recharge and water withdrawals, impacting water availability in the state. Reductions in the aquifer levels have been documented, impacting hydrologically connected springs, lakes, wetlands, and rivers [10]. A large portion of the state is currently designated as water resource caution areas [7] (Figure 1). Sea-level rise and continued population growth are expected to further tighten Florida's constrained groundwater and surface water supplies in the coming decades [11,12]. To ensure sufficient water resources are available to meet existing and future needs, the state needs to plan for significant investments to diversify water supply options toward alternative water supplies such as reclaimed water and brackish groundwater to ensure a sustainable water supply for the state [12,13].

Planning for sustainability is a complex task because of uncertainties around the water demand, water supply, and investment costs needed to balance water demand and supply (e.g., costs to increase water supply or costs to reduce water demand per capita through water conservation). Fund and water managers both face a similar challenge. They both need to have reliable water systems to meet water demand while sources of investment and water vary randomly [14–19]. Several approaches have been used to deal with the water supply and demand uncertainties: scenario-based robust optimization (RO) [14,20,21], adaptive pathways (AP) [22–24], and real options analysis (ROA) [15,25,26]. In general, AP and RO approaches are ruled-based planning frameworks [15,22]. Thus, to take not perfectly known future conditions into account, researchers need to somehow quantify the uncertainties (e.g., through a probability distribution and an ensemble of realizations [26,27]) and then embed these probability distributions or realizations into the model. Similarly, ROA is considered to be impractical without pre-defined distributions or realizations [15,27,28]. The second challenge of using ROA is that the method is not well-suited for quantifying the trade-off between the return and risks associated with various portfolio compositions [17,27–29].

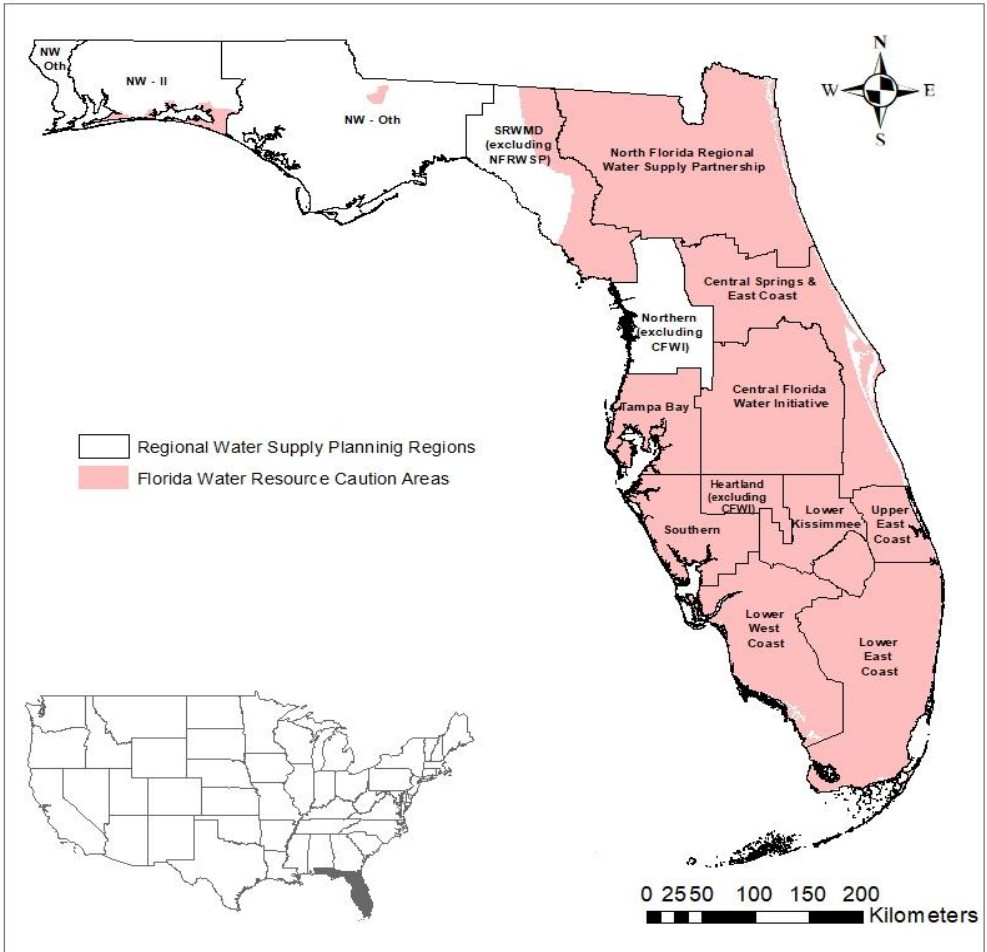

**Figure 1.** Florida's Water Resource Caution Areas and Water Supply Planning Regions. Note: Some planning regions cross WMD borders; these regional water supply plans were developed through collaboration by two or three WMDs. Source: Florida's Department of Environmental Protection (DEP), Office of Water Policy & Ecosystems Restoration [7].

Modern Portfolio Theory (MPT) approach has been widely used in finance to find optimal investment strategies for different financial assets under uncertainty. In the last two decades, however, several studies also applied MPT to water resources management, specifically, to optimize water resource protection and development given budget constraints and uncertainties in outcomes such as investment costs. For example, Ref. [17] showed that MPT can enhance the optimal investment strategy to achieve the maximum aggregated ecological benefit in a river catchment, given budget constraints and the uncertainty in investment benefits due to climate change. Ref. [30] used MPT to discuss flood mitigation investment options given risks associated with climate change and alternative land use development scenarios. Further, several studies argued for MPT use specifically for the choice of water supply infrastructure investment strategy. For example, Ref. [31] used this method to identify water supply diversification in a drought-prone area, given the uncertainty in water demand and reservoir water supply. Ref. [18] examined optimal water supply investments to reduce regional dependence on imported water with consideration of the uncertainty in hydrologic parameters. Similarly, Ref. [32] considered a region dependent on water import, but examined both water supply development and water demand management options, given a broad range of risks linked to water quality changes, climate change, energy price increases, and energy shortages. Other studies that rely on MPT to identify optimal water infrastructure investments are [20] (for decentralized water and wastewater investments) and [33] (for water distribution system investments).

For agricultural water supply, [27] used MPT to explore the joint use of managed aquifer recharge (MAR) and on-farm reservoirs with tail-water recovery (OFS-TWR) aimed at sustaining groundwater and agricultural income, given climate variability given a range of farmer risk preferences. [16] applied MPT to quantify the risk associated with alternative irrigation investment strategies in a water-stressed region. Overall, these studies argue that the major strengths of MPT are the systematic assessment of risks and the identification of investment options that are robust to uncertainty [30]. The method application, however, depends on data available to quantify the risk [17,29,30].

Despite an extensive literature on the use of MPT in natural resources and environmental management field, it remains unclear, though, the extent to which uncertainty of the costs of alternative water supply affect the total investment costs needed to meet the future water demand, and how water utility could reduce the financial risks associated with the uncertainty. In this paper, we perform the first large-scale analysis to compare different water supply options to improve water security and address water supply cost uncertainty in the context of Florida's natural resources, laws, and regulations. Much like the approach used in other U.S. states such as California and Texas [34,35], Florida's water agencies tend to rely on a scenario-based water supply investment planning, in which specific construction costs and capacities are assumed for future water supply projects. To enhance the scenario-based planning approach, this paper proposes to use the Modern Portfolio Theory (MPT) [36], which allows the selection of diversified water supply options to reduce the impact of capital cost uncertainty and manage the risk. We illustrate how using MPT can reduce the overall risk of the total investment portfolio in developing alternative water supply options. The approach relies on a principle that maximizes the expected investment returns for a given level of uncertainty (e.g., variance or standard deviation of the investment returns) or minimizes uncertainty for a given expected level of return. In addition, to account for the uncertainty of the projected expenditures, we use Chebyshev's Inequality, which allows estimating probability bounds based only on the mean and variance of an unidentified distribution [37,38]. We then apply this framework to estimate the range of expenditures needed to meet projected future water demand for each of Florida's regions by 2040.

This study contributes to the existing literature on MPT application to the choice of water supply investment strategies. Despite its somewhat exploratory nature, the first major contribution of this study is the insight into the extent to which diversification of water supply options reduces financial risk. Specifically, unlike previous studies (and particularly, [39]), this analysis examines the uncertainty associated with capital costs of alternative water supply investment options. The study confirms the findings of [16,31,32], and demonstrates that diversification of water supply options increases the resilience of water supply systems by reducing their financial risks. The second contribution of this study is that this study proposes a new method of accounting for cost uncertainty that can inform the process of policy and financial planning. While it is ideal to know the distribution of the costs when inferring the upper and lower bounds of the costs (e.g., 95% interval), such analysis typically requires an extensive dataset to identify the type of the distribution, which is rarely available to the researchers or water supply planners. A key strength of this study is the use of Chebyshev's Inequality to account for the capital costs uncertainty. The use of Chebyshev's Inequality does not require an assumption about the type of distribution of the costs that might follow [37,38]. To the best of our knowledge, this study has demonstrated, for the first time, that one can use Chebyshev's Inequality to quantify the uncertainty of the investments needed to meet rising water demand with limited data. The third contribution of this study is the introduction of a probabilistic-based approach to quantify the uncertainty of the investment costs to meet future water demand. Previous studies often provide estimations of the costs using descriptive statistical techniques [40,41], optimization-based [14,15,17,22,28], or econometric approach [39], and overlook the uncertainty of the costs and the extent to which the uncertainty affects the estimations [15,42].

We show that the total water use in Florida is projected to increase by 1405 MCM per year (+15.9%) by 2040, driven primarily by urbanization. Cumulative expenditures for the additional water supplies to meet future water demands are projected at the range of USD 1.11–1.87 billion, with an average of approximately USD 1.49 billion when median capital costs of alternative water supply projects are used. The results from Chebyshev's Inequality analysis indicate that projected expenditures can be well above USD 2.43 billion, with the range between USD 1.65 and USD 3.21 given the uncertainty about the project costs. Using MPT to determine the share of specific project types in the overall future water supply mix can reduce the standard deviation of capital cost per one unit of capacity by 74%, increasing confidence in water supply investment budgets. This study highlights the need for developing more flexible funding strategies at local, regional, and state levels to finance additional water supply infrastructure and to withstand the uncertainty around the projected expenditures. In addition, the results call for a more cost-effective combination of alternative water supply and demand management options to meet the water needed for the state in the future.

The remaining part of the paper proceeds as follows: the next section describes the data and empirical approach used in this study. The third section provides the results for the future water demand, expenditures needed to meet the demand, and an illustration of how MPT can be used to develop a more efficient combination of alternative water supply options. This section also provides the implications of the findings for future research in this area. Finally, the conclusion gives a summary and critique of the findings.

## 2. Materials and Methods

### 2.1. Study Area

The US state of Florida has largely relied on groundwater for water supply, but localized groundwater depletion and saltwater intrusion lead to withdrawal quantities from aquifers being capped in many regions across the state [11–13]. Brackish groundwater aquifers are also considered a finite resource due to limited recharge from the surface to this deeper zone [43].

Regulatory pressures on Florida's five water management districts (WMDs) have increased to ensure a sustainable water supply for Floridians. Florida Statutes require the WMDs to develop regional water supply plans in the regions where existing water sources are not adequate for all existing and future reasonable-beneficial uses and natural systems. These 20-year plans are required to identify sustainable water supply options and potential projects to meet future demands while protecting, conserving, and restoring water resources. The plans also include demand management strategies and water supply options to ensure sufficient water is available to meet the water supply needs of existing and future reasonable-beneficial uses for a 1-in-10-year drought event and to avoid adverse effects of competition for water supplies. To protect and restore natural systems, regional water supply plans also include projects from recovery and prevention strategies (RPS). RPSs are developed by WMDs for water bodies with flows or levels below the adopted minimum flows and minimum water levels (MFLs). RPSs are also developed for water bodies that are projected to fall below MFLs within 20 years. MFLs are defined as the limit at which further withdrawals would be significantly harmful to the water resources or ecology of the area (Chapters 373.709 and 373.042, Florida Statutes).

Florida's Department of Environmental Protection (DEP) provides the Governor and Florida Legislature with an annual status summary of water supply planning activities in each WMD. There are nineteen mutually exclusive water supply planning regions. To streamline the presentation of water supply planning activities in the WMDs, DEP combines six water supply planning regions located in the Northwest Water Management District, reducing the number of regions statewide from nineteen to fourteen in its reporting (Table 1). Regional water supply plans (RWSPs) are developed by the WMDs with consultations and feedback from stakeholders, and of the fourteen planning regions, they provide detailed descriptions of projected water demand, existing water supplies, as well as a range of

potential project options to meet the water demand, while protecting and restoring natural systems. Each plan has comprehensive information on the status of water resources within its boundary and how it can meet the future water demand considering the dynamics of natural, social, and demographic factors.

**Table 1.** Water Supply Planning Regions Considered in this Study.

| Water Management District | Water Supply Planning Region | Abbreviation | Water Supply Planning Document Referenced in [2] [a] |
|---|---|---|---|
| Northwest Florida Water Management District (NWFWMD) | I<br>III [a]<br>IV<br>V [b]<br>VI<br>VII | NW–Oth | 2018 Water Supply Assessment Update [2] |
| | II | NW–II | 2019 Region II Regional Water Supply Plan [2] [c] |
| Suwannee River Water Management District (SRWMD) | Area outside NFRWSP | SR–West | Water Supply Assessment 2015–2035 [2] |
| St. Johns River Water Management District (SJRWMD) | Central Springs and East Coast (Region 2, formerly Regions 2, 4, and 5) | SJR–CSEC | Under Development [2] [d] |
| Southwest Florida Water Management District (SWFWMD) | Northern Planning Region (partially in Central Florida Water Initiative) [e] | SW–N [e] | 2020 Regional Water Supply Plan; partially in CFWI Regional Water Supply Plan 2020 [2,4] |
| | Tampa Bay Planning Region | SW–TB | 2020 Regional Water Supply Plan [4] |
| | Heartland Planning Region (partially in Central Florida Water Initiative) [e] | SW–H [e] | 2020 Regional Water Supply Plan; partially in CFWI Regional Water Supply Plan 2020 [2] |
| | Southern Planning Region | SW–S | 2020 Regional Water Supply Plan |
| South Florida Water Management District (SFWMD) | Lower Kissimmee Basin | SF–LKB | Regional Water Supply Plan Update (2019) [2] |
| | Upper East Coast | SF–UEC | Regional Water Supply Plan Update (2016) [2,5] |
| | Lower East Coast | SF–LEC | Regional Water Supply Plan Update (2018) [2,6] |
| | Lower West Coast | SF–LWC | Regional Water Supply Plan Update (2017) [2] |
| SRWMD and SJRWMD | North Florida Regional Water Supply Partnership | NFRWSP | NFRWSP Regional Water Supply Plan [2] |
| SJRWMD, SWFWMD, and SFWMD | Central Florida Water Initiative | CFWI | CFWI Regional Water Supply Plan 2020 [2] |

Note: [a] The RWSP for Region III was first approved in 2008 and updated in 2014. This plan was discontinued in December 2018. [b] The Region V RWSP was approved in 2007 and discontinued in 2014. [c] The 2018 WSA is incorporated by reference, with the 2018 WSA containing the technical data, modeling tools, and methods used to develop the 2019 RWSP. [d] The demand estimates and projections are available in [2]. The draft RWSP was completed in July 2021. [e] In this report, the portion of the region outside the Central Florida Water Initiative is mentioned, with the abbreviations SW–N (for the Northern Region) and SW–H (for the Heartland Region).

It is widely recognized that significant investments will be required to ensure a sufficient water supply in the state. For example, a recent report to Florida Legislature estimates the future expenses for additional water supply to meet future water demand by 2040 would be approximately USD 1.5 billion [44]. Water supply investment planning is a priority at local, regional, and state levels. To our knowledge, none of the existing reports explicitly account for project cost uncertainty in regional investment planning.

## 2.2. Projecting Future Water Demand

In this study, the projection of total expenditures needed to meet the future water demand to 2040 relies on three sources of information: projected future water demand, inferred future water supply, and estimated capital costs of water supply alternatives. Projections of water demand here differ in important ways from what the economics literature would typically refer to as water demand. Modeling water demand in the traditional economic sense would involve modeling demand functions and responses to price signals and other demand characteristics. Estimating such demand functions for 14 water supply regions considered in this study would be not only a challenging task but unrealistic because of the need for a large amount of data across the entire state. To come up with plausible water demand projections, we compile water use projections from Florida's water management districts (WMDs) and Florida's Statewide Agricultural Irrigation Demand (FSAID). The WMDs provide water demand projections for six water use categories: public supply (PS), domestic self-supply (DSS), agriculture (AG), landscape/recreational (L/R), commercial/industrial/institutional (CII), and power generation (PS). For most regions, 2040 water demand projections are available, however, two regional water supply plans (RWSPs) (i.e., SR-West and NFRWSP) only project water demand to 2035. For these two regions, we extend the projections to 2040 with linear trends. Herein, we use the projections of water use from WMDs and Florida's Department of Agriculture and Consumer Services (FDACs), and consider these projections are reasonable representations of future water demands. A more detailed explanation of the methods used to project the water demand for PS, DSS, L/R, CII, and AG can be found in the 2020 Annual Status Report on Regional Water Supply Planning, Florida Department of Environmental Protection (DEP) [6], Annual Assessment of Florida's Water Resources: Supply, Demand, Florida Office of Economic and Demographic Research (EDR) [44], and FSAID [45]. We summarize the methods used to project the water demand in Appendix A.

## 2.3. Inferring Current Water Supply and Water Shortage

The water supply in a particular year depends on many factors, which poses a challenge for a reliable prediction of water supply limits for a planning horizon. The dynamic nature of hydrogeology and water quality do not easily lend themselves to calculating a specific static water supply [46]. For example, Tampa Bay water is one of the largest water utilities in Florida, providing water for about 2.5 million people [47]. Their projected water demand in 2035 varies by almost 20 percent between the low and the upper estimates (i.e., between 322 and 386 MCM per year), complicating the analysis of potential water supply needs and sources [46]. At the same time, WMDs are required to include analyses of future water demand and supply limits for the regions covered by their RWSPs.

In this study, we infer the current water supply from the demand projections, and water needs for PS, DSS, L/R, and CII reported by the WMDs in their RWSPs, and AG from FSAID-8 [45]. Specifically, we use two pieces of information: first, projected regional water demand, and second, regional water needs at the end of the planning horizon (which we also refer to as water supply shortages). As shown in Figure 2, the inferred water supply can be estimated as the difference between projected water demand and water supply shortage reported for the end of the planning horizon. The inferred water supply does not change over time, reflecting the fact that bringing a new supply project may take up to ten years or longer from project inception to full implementation. The inferred water

supply serves as the best proxy of the current water supply, which is needed to estimate the expenditure, and total investment needed to meet the future water demand.

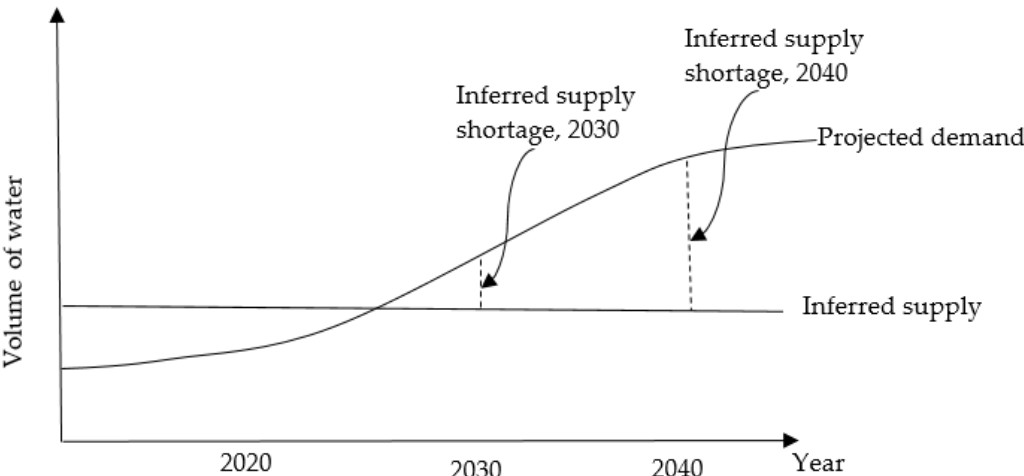

**Figure 2.** Schematic Illustration of Inferred Water Supply Shortage Calculations.

Figure 2 also illustrates how water supply shortages are projected for each region, using the inferred water supply and projected water demand. Herein, projected water supply shortages at any point in time are equal to projected water demand in RWSPs (from WMDs) minus inferred supply. Thus, this approach results in projected water supply shortages equal to the water need reporting in RWSPs from WMDs.

### 2.4. Estimating the Capital Costs of Water Supply Alternatives

To estimate the capital costs of water supply alternatives, we utilize the information about the project options identified in the RWSPs, including projects from Recovery or Prevention Strategies (RPSs) (Table 2). We also use the information about the projects implemented and funded by the WMDs or state agencies in the past, and the projects currently being designed or constructed (and funded or co-funded by agencies) [6].

Florida's Water Management Districts are required by Section 373.709, Florida Statutes, to compile a list of project options for water supply development and water resource development for each water supply planning region. Each project option included in the list contains multiple attributes such as location, type, capacity, and total cost. The project list is included in the project summary in [6], which is the most comprehensive statewide dataset of the Florida water supply development and water resources development projects. Currently, the list primarily includes projects that are eligible for districts or state cost-share funding programs, and it summarizes completed projects, projects currently in design and construction, canceled and on-hold projects, and potential future projects that may be implemented to address regional water needs. The project list currently includes 1694 project items. Removing canceled projects, 1629 project items remain for further analysis. The remaining projects can be classified into four general categories: additional water supply projects, water demand management and conservation projects, natural system projects, and others (Table 2). The final number of projects used in this study excludes 78 projects categorized as other. Following the approach adopted by WMDs, we report water conservation as a project type, comparable with water supply projects. We recognize, however, that water conservation impacts water use, as opposed to water supplies. We convert the project cost to the USD 2021 value using Engineering News Record Construction Cost Index History [48]. The estimated expenditures for reclaimed water projects account for the beneficial offset being only 0.55 of the actual project capacity [49].

**Table 2.** General project categories used in this study.

| Project Category | Project Description | Number of Projects * |
|---|---|---|
| Additional water supply to meet growing demand | Projects in the regions with positive 2040 inferred supply shortages, given that the projects are not associated with any MFL RPS. Specifically, the following project types are considered:<br><br>• Reclaimed Water (for potable offset)<br>• Brackish Groundwater<br>• Surface Water<br>• Surface Water Storage<br>• Groundwater Recharge<br>• Aquifer Storage and Recovery (ASR)<br>• Stormwater<br>• Other Project Type<br>• Other Non-Traditional Source<br>• Desalination<br>• Distribution/Transmission Capacity | 960 |
| Water demand management and conservation | • PS and CII Conservation<br>• Agricultural Conservation | 570 |
| Water for natural systems | • All projects that are not yet completed and that are associated with specific MFL RPS<br>• Reclaimed water projects for groundwater recharge or natural system restoration, if the project status is listed as in design, in construction/underway, or on hold<br>• All project types if the projects are in the regions with no inferred shortage, if the project status is in design, in construction/underway, or on hold | 165 |
| Other | • Flood Control Works<br>• Data Collection and Evaluation | 78 |

\* Note: the total is greater than the total number of the projects considered (1629) in the dataset since some projects fall into more than one category.

*2.5. Applying Chebyshev's Inequality to Infer the Bounds for the Capital Costs of Water Supply Alternatives*

The project list shows that the capital costs per unit of capacity of water supply alternatives vary substantially. Our analyses indicate that these costs are not normally distributed (i.e., the mean differs from the median). In this case, using the mean or median of the costs might greatly under or over-estimate the expenditures. We know very little about the distributions of the costs for each project type, but we would want to make plausible inferences for the upper and lower bounds of these costs. We apply Chebyshev's Inequality to accomplish this task.

If $X$ is a non-negative random variable, with an expected value of $E(X) = \mu$ and variance $Var(X) = \sigma^2$, then, for every real number $a > 0$, Markov's Inequality (Equation (1)) shows the probability of random value $X$ is equal to or smaller than the expected value divided by the real number, $a$, as shown below:

$$P(X \geq a) \leq \frac{E(X)}{a} \tag{1}$$

If a random variable is defined as $Y = (X - \mu)^2$, then $E[Y] = Var(X) = \sigma^2$. Applying Markov's Inequality to $Y$ with $a = k^2$ gives us Chebyshev's Inequality as:

$$P\left(Y \geq k^2\right) \leq \frac{E(Y)}{k^2} \tag{2}$$

$$P\left((X - \mu)^2 \geq k^2\right) \leq \frac{Var(X)}{k^2} \text{ or} \tag{3}$$

$$P(|X - \mu| \geq k) \leq \frac{\sigma^2}{k^2} \tag{4}$$

When $k = d\sigma$ and $d$ is a non-negative number, Chebyshev's Inequality, Equation (4), can be written as:

$$P(|X - \mu| \geq d\sigma) \leq \frac{1}{d^2} \tag{5}$$

If $X$ is a capital cost of a water supply project randomly selected from the project sample, and if k is the upper bounds of the costs, then $P(|X - \mu| \geq k) \leq \frac{\sigma^2}{k^2}$ with $\mu$ and $\sigma^2$ being the mean and variance of the costs, respectively. In this study, we would want to find the bounds of capital costs, defining the pseudo 95% confidence interval. That is, we want to know the costs, $C = [C_u, C_l]$, which make their probabilities fall within the pseudo 95% confidence interval, $0.05 \leq P(X \geq C) \leq 0.95$. The costs with probability can be expressed as:

$$P(X \geq C) \leq \frac{1}{d^2} \tag{6}$$

With $C = \mu + d\sigma$, we can rewrite Equation (6) as:

$$P(X \geq C) \leq \frac{1}{\left(\frac{C-\mu}{\sigma}\right)^2} \tag{7}$$

Given the probabilities (e.g., 0.05 and 0.95), mean, $\mu$, and standard deviation, $\sigma$, of the costs, we solve for the $C$.

To account for the suitability of project type in each water supply planning region, we follow [50,51] and from the project list in [6], we retain only project types ranked as "highly" or "moderately likely" to be viable. Thus, the final bounds of the costs are the combined cost bounds for the projects ranked as highly or moderately likely.

### 2.6. Applying Modern Portfolio Theory to Select a Mix of Water Supply Alternatives

Modern Portfolio Theory (MPT) provides a mathematical framework where investors choose optimal portfolios based on risk and return. One of the key principles of MPT is that one can reduce the risk through diversification. The MPT can be formulated as (1) minimizing the risk (standard deviation/variance of return) given a specified return or (2) maximizing the return given a specified risk.

If we have $n$ choices of water supply project type, $X_i$, where $i = 1, \ldots n$, we build a portfolio, $F$, with expected return as:

$$E(F) = \sum_{i=1}^{n} w_i R(X_i) \tag{8}$$

where,

$R(X_i)$ is the return of project type $X_i$

$w_i$ is the proportion of project type $X_i$ in the portfolio, where $\sum_{i=1}^{n} w_i = 1$

If $\theta$ denotes the covariance matrix for options $X_i$, the variance of the portfolio can be expressed as:

$$\sigma_F^2 = w^T \theta w \tag{9}$$

One can solve for $w_i$ by setting the first derivative of the variance equal to zero.

Consider two project-type scenarios with the proportion of project 1, $w_1$, and of project type 2, $w_2 = 1 - w_1$. Let the covariance of the two project types be $\sigma_{12}$. The variance of the portfolio is expressed as:

$$\sigma_F^2 = w_1^2 \sigma_1^2 + (1 - w_1)^2 \sigma_2^2 + 2w_1(1 - w_1)\sigma_{12} \tag{10}$$

where $\sigma_{12} = \rho_{12}\sigma_1\sigma_2$ is the covariance of $X_1$ and $X_2$. $\rho_{12}$ is the correlation between $X_1$ and $X_2$.

The 1st derivative of the variance is

$$\frac{\partial \sigma_F^2}{\partial w_1} = 2w_1\sigma_1^2 - 2\sigma_2^2 + 2w_1\sigma_2^2 + 2\rho_{12}\sigma_1\sigma_2 - 4w_1\rho_{12}\sigma_1\sigma_2 \tag{11}$$

Set the derivative equal to zero, we have

$$2w_1\sigma_2^2 + 2w_1\sigma_1^2 - 4w_1\rho_{12}\sigma_1\sigma_2 = 2\sigma_2^2 - 2\rho_{12}\sigma_1\sigma_2 \tag{12}$$

Then, the weight of $X_1$, $w_1$, is equal to

$$w_1 = \frac{\sigma_2^2 - \rho_{12}\sigma_1\sigma_2}{\sigma_1^2 + \sigma_2^2 - 2\rho_{12}\sigma_1\sigma_2} \tag{13}$$

When there are more than two alternative investments, finding the solution with an analytical approach becomes less tractable. One can use a mathematical optimization solver (e.g., Simplex LP in Microsoft Excel, GAMS, and Matlab) to find the optimal solution for the weights. The optimization problem would be to minimize the risk (i.e., variance or standard deviation), subject to nonnegative and additive constraints for the weights, and a selected level of return. Alternatively, one can solve for the optimal weights by maximizing the return given a level of risk.

In this paper, we minimize the variance of the capital costs given various levels of the costs to build an efficient frontier. We construct an efficient frontier to demonstrate how MPT can be used to find a mix of water supply alternatives with the lowest risk level given a return. Portfolios on the efficient frontier are optimal in both offering maximum expected return for a level of risk and minimal risk for a given level of return. It is ideal to build an efficient frontier using the total net return per unit of capacity for water supply projects. The total net returns of alternative water supply projects are highly speculative because the total benefits and costs of water supply projects are often unavailable [29,30,44]. Herein, we use capital costs to make this efficient frontier with the assumption that the benefits per unit capacity are the same for any water supply project. This assumption makes the variance of the capital costs equal to the variance of the returns.

### 3. Results and Discussion

*3.1. Water Demand Projections*

Development water demand includes freshwater demand for Public Supply (PS), Domestic Self Supply (DSS), Landscape/Recreational (L/R), Commercial/Industrial/Institutional (CII), and Power Generation (PG). Among these water demand categories, PS water demand is by far the largest in Florida. PS water demand is typically projected by WMDs from water use per capita for five base years multiplied by projected population growth [11–13,44]. The PS water use projection thus assumes a static water use efficiency, not accounting for future possible technological changes and changes in urban development standards (such as reductions in the irrigated area). This method produces conservative demand projections, potentially providing for a "safety margin" in water supply development since the changes in water use behavior, technology, and building standards often lead to a reduction in water use per capita [44].

The water forecast for agricultural use comes from Florida's Statewide Agricultural Irrigation Demand (FSAID) [44,45]. The Florida Department of Agriculture and Consumer Services (FDACS) oversees the development of statewide agricultural water demand projections reported in FSAID. The forecast is based on a model fitted to 2007–2019 metered or reported permit-level agricultural water use data. In addition, the data used to develop the model include land use/land cover from WMDs, Consumptive Use Permit polygons from the WMDs, well locations, USDA's Cropland Data Layer data, USDA's National Agricultural Imagery Program (NAIP) aerial imagery, and Irrigated Areas layers from SJRWMD and SWFWMD [45].

Figure 3 shows the summary of the statewide water demand forecasts. The result is quite revealing in several ways. First, there is an expected 15.9% increase in total water use during 2020–2040, but the most substantial increase in the statewide water demand is projected from the development of water demand (which includes residential water demand). Agricultural water demand is expected to stabilize in the next 20 years while the development water demand is projected to increase by more than 22%, from 5493 MCM in 2020 to 6725 MCM in 2040. Second, the projected increase in water demand is highly correlated with population growth, indicating that urbanization is the main driver of the projected increase in water use. Third, despite the expected strong urbanization and population growth, the agricultural water demand is projected with only a slight increase, indicating that few changes can be expected to irrigated acres or irrigation rates within the state [44,45] despite an expectedly strong population growth [3] and thus food demand.

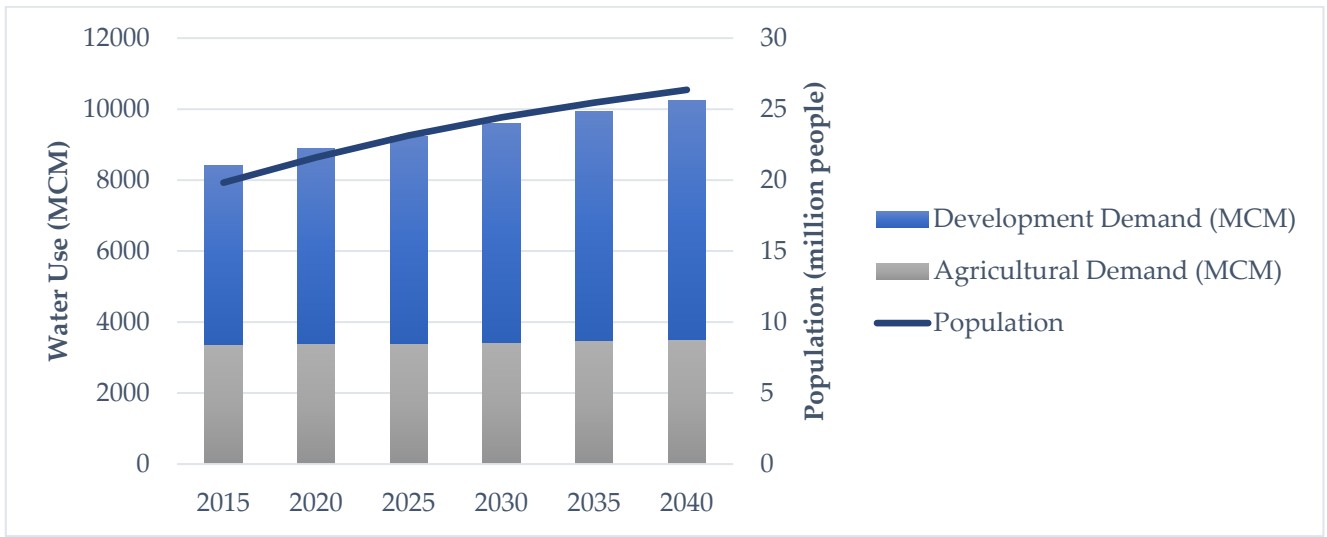

**Figure 3.** Annual water demand projections for development and agriculture. Development water demand includes water demand for Public Supply (PS), Domestic Self Supply (DSS), Landscape/Recreational (L/R), Commercial/Industrial/Institutional (CII), and Power Generation (PG) come from the 2020 Annual Status Report on Regional Water Supply Planning, Florida Department of Environmental Protection (DEP) [6]. Water demand for agriculture (AG) comes from Florida Statewide Agricultural Water Demand (FSAID) [45].

Figure 4 shows the water use and water shortage forecasts for 2020 and 2040 for each water supply planning region. The water use forecasts included in this table present the "status quo" scenario for the per capita water use, not accounting for possible future water use efficiency improvements or water conservation, or potential longer and more severe drought that could increase the water use. The forecast shows that South Florida (i.e., SF–UEC, SF–LEC, and SF–LWC) likely has the largest increase in water use, followed by CFWI and NFRWSP. However, regarding the water shortage, NFRWSP is projected to need the most alternative water supplies and/or conservation to meet future demands, followed by CFWI and SJR–CSEC (see Table 3).

*3.2. Existing Water Supplies, Currently Implemented Projects, and Remaining Potential Water (Needs) Shortages*

In general, the RWSPs indicate that the current water supply sources are unlikely to meet the future water demand. Table 3 identifies the water needs at the end of the planning horizon (aka potential water supply shortages), calculated from the existing water sources and projected future water use reported by WMDs. The unmeet future water demand is expected to be met through alternative water supplies, as well as water conservation [6,44].

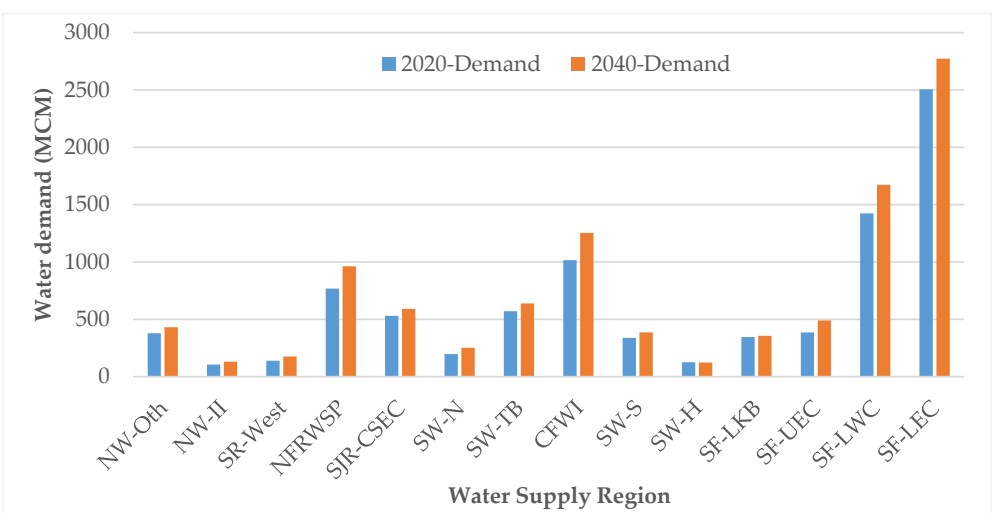

**Figure 4.** Total Annual Demand Projections and Potential Water Needed by Water Supply Planning Regions. Note: Two regions, SR-West and NFRWSP have water projections to 2035 only. Linear extrapolations of water use are applied to project water demands to 2040. Source: development water demands (i.e., PS, DSS, CII, L/R, and PG) comes from the 2020 Annual Status Report on Regional Water Supply Planning, Florida Department of Environmental Protection (DEP) [6]. Agricultural water demand (i.e., irrigation, livestock, aquaculture, and freeze protection water demand) comes from Florida Statewide Agricultural Water Demand (FSAID) [32]. Note: Two regions, SR-West and NFRWSP have water projections to 2035 only. Linear extrapolations of water use are applied to project water demands to 2040.

**Table 3.** Analysis of the Recently Completed Projects and Projects in Construction, in Design, and On Hold, by Regions with Water Shortages *.

| Planning Regions | Water Needed (aka Potential Inferred Supply Shortage by 2040, MCM) | Water by the Projects in Design, Construction, and on Hold, MCM | Remaining Potential Inferred Supply Shortage by 2040, MCM ** | Project Expenditures by the Projects in Design, Construction, and on Hold (million, USD 2021) |
|---|---|---|---|---|
| (1) | (2) | (3) | (4) = (2) − (3) | (5) |
| NWF–II | 6.91 | 6.92 | - | USD 21.16 |
| SR–West | 7.17 | 2.75 | 4.42 | USD 5.01 |
| SJR–CSEC | 70.60 | 35.69 | 34.91 | USD 156.07 |
| SW–N *** | 15.96 | 0.62 | 15.34 | USD 30.46 |
| SF–UEC | 5.18 | 282.08 | - | USD 11.64 |
| SF–LEC | 68.46 | 143.92 | - | USD 32.62 |
| SF–LWC | 12.81 | 78.02 | - | USD 22.13 |
| NFRWSP | 195.23 | 14.18 | 181.05 | USD 28.48 |
| CFWI | 131.26 | 111.43 | 19.83 | USD 339.61 |
| Statewide (sum of regions) | 513.58 | 675.61 | 255.55 | 647.18 |

Note: * The table focuses on the regions with "Water Need" identified in the 2020 Annual Status Report on Regional Water Supply Planning, Florida Department of Environmental Protection (DEP) [6]. The "Water Need" defines as expected water shortages that need to be met by alternative water supply options. Projects considered to be for natural system restoration are excluded. These are the projects associated with MFL RPS, reclaimed water (for groundwater recharge or natural system restoration), and most of the projects described as restoration (in the "Project Description" field). We compute these values based on the capacity of the projects listed in the 2020 Annual Status Report on Regional Water Supply Planning, Florida Department of Environmental Protection (DEP) [6]. ** Negative values of the inferred shortage are not reported. *** Excluding CFWI.

Since [6] identifies several current projects to address future water needs. Therefore, the remaining potential inferred supply shortage is equal to the "Water Needed" by 2040 minus the water that will be supplied by the projects in design, construction, and on hold (Table 3). The last column in Table 3 indicates the total expenditures committed to the projects in design, construction, and on hold. In general, the results presented in Table 3 show that statewide expenditures by the projects in design, construction, and on-hold are expected to be equal to USD 647.48 million. A major portion of the expenditures comes from CFWI accounting for 52% of the total expenditures, but the bulk of the remaining potential inferred water supply shortage comes from NFRWSP, followed by SJR–CSEC and CFWI. It is not surprising that NFRWSP has the largest water supply shortage because this is a sizable region, which overlays large portions of two out of five WMDs in the state. Table 3 also reveals that the total expenditures committed to projects in design, construction, and on hold in NFRWSP are relatively small (USD 28.48 million) compared to CFWI (USD 339.61 million) and SJR–CSEC (USD 156.07 million).

*3.3. Water Supply Alternatives to Address Water Needs*

Figure 5 illustrates the range of capital cost estimates for alternative water supply projects at the statewide level. Variability in project type, design, and location likely results in a wide range of costs [6,39]. At the low end of the costs range, stormwater, groundwater recharge, and surface water with a median cost of fewer than one million dollars per MCM of annual project. These projects reflect those requiring additional infrastructure to convey water to project sites such as stormwater capture and groundwater recharge. Reclaimed water projects have the highest median cost, and also the widest range of cost. The median cost of reclaimed water is approximately five million dollars per MCM per year for portable reuse projects while less than four million dollars for non-portable projects. Water recycling for non-portable reuse projects is typically less expensive than potable reuse because non-potable reclaimed water requires less treatment than potable projects. As noted earlier, these costs are capital costs and represented a single phase of these projects (i.e., construction). These estimates likely underestimate total costs for alternative water supply development, which include additional planning design, permitting costs, transmission, rehabilitation, or replacement of existing facilities and systems (let alone operation and maintenance costs).

Table 4 shows the median, lower bound, and upper bound of the estimated project costs per one unit of capacity calculated from the project list [6]. The estimated lower and upper bounds for the costs are estimated using Chebyshev's inequality while median costs are the midpoint of the costs. Herein, we only present the costs for the three project types that are considered highly and moderately likely sources to meet future demands [6,51]. Columns (2)–(4) show the median, lower and upper bounds of the costs. The last four columns of the table indicate the final bounds of the costs used to estimate the expenditures. For each water planning region, a less expensive expenditure means the least expensive projected costs per unit of capacity among alternative water supply options considered. Similarly, a more expensive cost shows the most expensive water supply option for that region. For example, when Chebyshev's inequality is applied to estimate the bounds of the costs, CFWI has the projected less and more expensive costs for USD 6.30 and USD 21.06 million per MCM, respectively. The value of USD 6.30 million per MCM is the minimum of USD 6.30 and USD 8.03 million MCM. Likewise, USD 21.06 is the result of the maximum of USD 18.82 and USD 21.06. These estimated bounds of the costs account for the uncertainty of the costs. Thus, these bounds are considerably higher than that when median costs are used. CFWI has the projected less and more expensive expenditure for USD 0.89 and USD 3.50 million per MCM, respectively when median costs are used. However, the range should be USD 6.30–21.06 when Chebyshev's inequality is used to account for the uncertainty of the costs. In addition, as shown in Table 4, there is a considerable variation in the costs among the regions. For example, reclaimed water projects in SJR–CSEC has a median cost of USD 6.72 million per MCM per year while it is estimated to be equal to USD

10.62 million in SW–N. The cost is expected to be much lower than that, approximately USD 4.05 million in NFRWSP and CFWI.

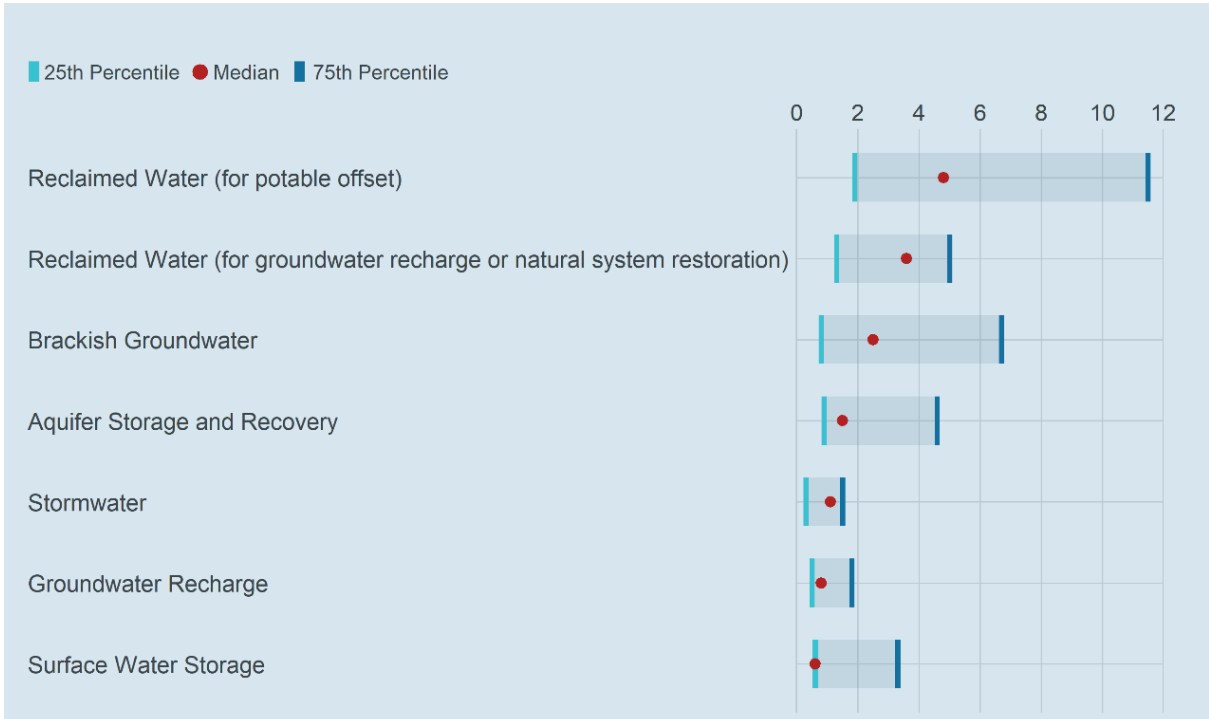

**Figure 5.** Estimated Project Expenditures per Annual Unit of Capacity for Alternative Water Supply Projects (million USD 2021 per MCM). Data source: 2020 Annual Status Report on Regional Water Supply Planning, Florida Department of Environmental Protection (DEP) [6].

**Table 4.** Estimated Project Expenditures per Unit of Capacity (million USD 2021 per MCM).

| Planning Regions | Median, Lower and Upper Bounds * | | | Using Median | | Using Chebyshev's Inequality | |
|---|---|---|---|---|---|---|---|
| | Brackish Groundwater | Groundwater Recharge | Reclaimed Water | Less Expensive | More Expensive | Less Expensive | More Expensive |
| (1) | (2) | (3) | (4) | (5) | (6) | (7) | (8) |
| SR–West | | | 8.88-8.88-8.88 ** | 8.88 | 8.88 | 8.88 | 8.88 |
| SJR–CSEC | 4.05-4.05-4.05 ** | | 6.72-10.50-26.78 | 4.05 | 6.72 | 4.05 | 26.78 |
| SW–N *** | | | 10.63-15.05-33.29 | 10.63 | 10.63 | 15.05 | 33.29 |
| NFRWSP | | 0.56-2.55-7.60 | 3.97-9.77-26.42 | 0.56 | 3.97 | 2.55 | 26.42 |
| CFWI | 0.89-6.30-18.82 | | 3.50-8.03-21.06 | 0.89 | 3.50 | 6.30 | 21.06 |

Notes: * Values in this table assume the median project capacity. For reclaimed water projects, the beneficial offset is assumed to be 55 percent of each project's capacity [6,49]. The numbers presented in the table are the median, lower bound, and upper bound of the cost, respectively. ** Sample sizes are too small for computing up and lower bounds for the region. Thus, we use the median as a proxy of the bounds to allow us to infer the expenditures for the region. *** Excluding CFWI.

Regarding the estimated bounds of the costs, a project type with smaller median costs does not necessarily have a tighter cost bound because the bounds reflect the uncertainty of the costs, which largely depend on the variance, and to a lesser extent, the mean of the costs. The regional cost variance within the dataset is reflected in the estimated cost bounds in Table 4. For instance, the estimated median costs for reclaimed water projects in SJR–CSEC (USD 6.72 million) are much higher than that in NFRWSP (USD 3.97 million), but their bounds are comparable, with USD 10.50–26.78 million in SJR–CSEC and USD 9.77–26.42 million in NFRWSP. This table also reveals that the distributions of these costs are

unlikely to be normal, showing positive skew, as their means are greater than their medians (Figure 5). Another indication of positive skew distributions of the costs is that lower bounds are greater than the medians for all water planning regions considered. This finding implies that using median costs for estimating the expenditures likely underestimates the expenditure needed to meet future water demand in the state.

Overall, we find that the costs of alternative supply options can be highly varied within the same region and methods used (see Table 4). Accounting for uncertainty leads to a much wider range of costs than that in scenario-based planning with the median cost analysis. Groundwater recharge projects are among the least expensive alternative water supplies, which is consistent with the previous estimates (see Figure 5, which is based on agency reports [6,38]). However, the option does not seem to be used widely, and its suitability may depend on the aquifer characteristics that vary widely across Florida. Currently, groundwater recharge is widely used in NFRWSP only [6]. The most expensive alternative water supply is reclaimed water, but it is also the most widely used one because of the potentially broad availability of reclaimed water in growing urban areas. As noted, the costs used in this analysis are capital costs, which might not be reflective of the total costs over the life of the projects examined in this study.

### 3.4. Total Investments Needed to Meet Projected Water Use in Florida

In this section, we compare the total estimated investment or expenditure to meet future water demand using the two methods. First, the expenditures are estimated using the median of the estimated project expenditures per one unit of capacity calculated from the project list coming from [6]. Second, we estimate the total investment using the project uncertainty analysis based on Chebyshev's inequality.

Table 5 shows the projected expenditures based on the median of the expenditures per one unit of capacity calculated from the project list [6]. For each planning region, the values in column (2) are equal to the projected less expensive expenditure per unit of capacity presented in Table 4 multiplied by the remaining potential inferred supply shortage by 2040 (column (4) in Table 4). Similarly, the values in column (3) are the product of the multiplication of the projected more expensive expenditure per unit of capacity presented in Table 4 multiplied by the remaining potential inferred supply shortage by 2040. For example, SJR–CSEC has projected less and more expensive expenditures of USD 141.51 and USD 234.51 million, respectively because the potential inferred water supply shortage is 34.91 MCM and the less and more expensive expenditures are USD 4.05 and USD 6.72 million per MCM, respectively. As can be seen, the total projected expenditures needed to increase water supply are in the range of USD 1109.57–1875.56 million, with an average of USD 1490.57 million. As noted earlier, the distribution of the capital costs is likely not normal, but positively skewed. Using median costs might result in underestimated expenditures. However, if the projected demand is a "conservative" projection, one might argue that actual future water demand should be lower than the demand presented in this table, and thus the answer to the question about whether the estimated expenditure is "conservative" is underdetermined. Regardless of the distribution of the costs, our projected expenditure agrees with [26]. The authors relied on an econometric model to show that the projected statewide expenditure was USD 1752.85 million [26].

Table 6 shows the estimates of projected expenditures needed to meet future water demands with estimated unit cost using Chebyshev's inequality. As can be seen, there is a large uncertainty around the projected expenditures needed to meet future water demand by 2040. The uncertainty appears to increase the expenditure estimates compared to the estimates presented in Table 5 substantially, by approximately one billion dollars by 2040. The lower and upper bounds of the total projected expenditures are equal to USD 1646.28 and USD 3212.28 million, respectively compared to the range of USD 1109.57–1875.56 million when median capital costs are used to project the expenditures (see Table 5). A possible implication of this is that flexible funding strategies at the local, regional, and state levels

are needed to withstand the uncertainty around the investments needed to meet future water demand statewide.

**Table 5.** Projected Expenditures for the Additional Water Supply using Median Project Expenditures per Unit of Capacity.

| Planning Regions | "Project Total" to Meet Remaining Inferred Shortage (Million, USD 2021) | | Project Expenditures by the Projects in Design, Construction, and On Hold (Million, USD 2021) | Total Forecasted Expenditure to Meet 2040 Inferred Supply Shortage (Million USD 2021) | | |
|---|---|---|---|---|---|---|
| | Less Expensive | More Expensive | | Less Expensive | More Expensive | Average |
| (1) | (2) | (3) | (4) | (5) = (2) + (4) | (6) = (3) + (4) | ((6) + (7))/2 |
| NWF–II | - | - | 21.16 | 21.16 | 21.16 | 21.16 |
| SR–West | 39.26 | 39.26 | 5.01 | 44.27 | 44.27 | 44.27 |
| SJR–CSEC | 141.51 | 234.51 | 156.07 | 297.58 | 390.58 | 344.08 |
| SW–N * | 163.06 | 163.06 | 30.46 | 193.52 | 193.52 | 193.52 |
| SF–UEC | - | - | 11.64 | 11.64 | 11.64 | 11.64 |
| SF–LEC | - | - | 32.62 | 32.62 | 32.62 | 32.62 |
| SF–LWC | - | - | 22.13 | 22.13 | 22.13 | 22.13 |
| NFRWSP | 100.90 | 718.10 | 28.48 | 129.38 | 746.58 | 437.98 |
| CFWI | 17.65 | 69.45 | 339.61 | 357.26 | 409.06 | 383.16 |
| Total | 462.39 | 1224.38 | 647.18 | 1109.57 | 1871.56 | 1490.57 |

Note: the values in column (4) are the project expenditures by the projects in design, construction, and on-hold presented in column (5), Table 4. * Excluding CFWI.

Some of the challenges for this analysis relate specifically to (1) the method used to estimate the bounds of the capital costs, (2) data sources and limitations regarding how the costs estimated, and (3) the accuracy of the projected future water demand. In this study, we use Chebyshev's Inequality to compute the bounds. Chebyshev's Inequality is known to provide "conservative," larger bounds. These estimated bounds intend to reflect the uncertainty around the projected expenditures, but the bounds do not explain the occurrence of events and/or factors that affect these bounds. Using Chebyshev's Inequality is particularly useful when we do not have adequate information to explain the variation of the costs [17,18]. Regarding the second issue, despite the data covering many projects in the state, there are some project types with only a few projects in selected regions. For example, there are only a few reclaimed water projects included in this analysis for the SR–West region. The small and insufficient sample size might lead to a large error in the estimates presented in Table 6. The third issue is related to water demand estimations. The projected expenditures presented in Tables 5 and 6 rely heavily on the accuracy of the projected future water demand coming from WMDs. As noted earlier, the future water demand projections likely represent the "conservative" estimates, which expectedly lead to higher projections of expenditures. It is worth noting that the projected expenditures needed in Tables 5 and 6 represent the capital costs of a single phase of the projects (i.e., project construction) typically funded by the state or water management districts/entities. This estimate likely underestimates total costs for alternative water supply development, which include additional planning design, permitting costs, transmission, rehabilitation, or replacement of existing facilities and systems (not to mention operation and maintenance costs). The state likely needs substantially higher than the projected expenditures to supply enough water for Floridians. According to [52], the state's drinking water utilities need approximately USD 21.88 billion to meet the future water demand by the end of 2034. In addition, this estimated expenditure also excludes the expenditure needed to meet the future water demand for ecosystems.

**Table 6.** Projected Expenditures for the Additional Water Supply using Chebyshev's inequality Bounds Project Expenditures.

| Planning Regions | "Project Total" to Meet Remaining Inferred Shortage (Million, USD 2021) | | Project Expenditures by the Projects in Design, Construction, and On Hold (Million, USD 2021) | Total Forecasted Expenditure to Meet 2040 Inferred Supply Shortage (Million USD 2021) | | |
|---|---|---|---|---|---|---|
| | Less Expensive | More Expensive | | Less Expensive | More Expensive | Average |
| (1) | (2) | (3) | (4) | (5) = (2) + (4) | (6) = (3) + (4) | ((6) + (7))/2 |
| NWF–II | - | - | 21.16 | 21.16 | 21.16 | 21.16 |
| SR–West | 39.26 | 39.26 | 5.01 | 44.27 | 44.27 | 44.27 |
| SJR–CSEC | 141.49 | 366.62 | 156.07 | 297.56 | 522.69 | 410.125 |
| SW–N * | 230.93 | 230.93 | 30.46 | 261.39 | 261.39 | 261.39 |
| SF–UEC | - | - | 11.64 | 11.64 | 11.64 | 11.64 |
| SF–LEC | - | - | 32.62 | 32.62 | 32.62 | 32.62 |
| SF–LWC | - | - | 22.13 | 22.13 | 22.13 | 22.13 |
| NFRWSP | 462.56 | 1768.99 | 28.48 | 491.04 | 1797.47 | 1144.255 |
| CFWI | 124.86 | 159.31 | 339.61 | 464.47 | 498.92 | 481.695 |
| Total | 999.10 | 2565.10 | 647.18 | 1646.28 | 3212.29 | 2429.29 |

Note: the values in column (4) are the project expenditures by the projects in design, construction, and on-hold presented in column (5), Table 4. * Excluding CFWI.

### 3.5. Costs of Water Efficiency Measures

Figure 6 illustrates the range of capital costs of AG, PS, and CII water conservation projects. In general, agricultural water conservation projects have similar median costs compared to PS and CII water conservation projects. However, the range of cost for agricultural water conservation projects is much larger than that of PS and CII water conservation projects, suggesting that this cost depends on project type and size, design, and location. In general, water efficiency measure projects likely have a lower cost compared, for example, to commonly used reclaimed water projects, but these projects likely cost more than other alternative water supply project types (Figure 5). While water conservation programs considered by the WMDs also require investments, many PS and CII water conservation projects are likely to be less expensive than reclaimed water supply projects in many water supply planning regions. Thus, increasing the adoption rate of urban water conservation would be a more efficient approach to meet the water demand in the regions where the costs of alternative water supply projects such as reclaimed water are high.

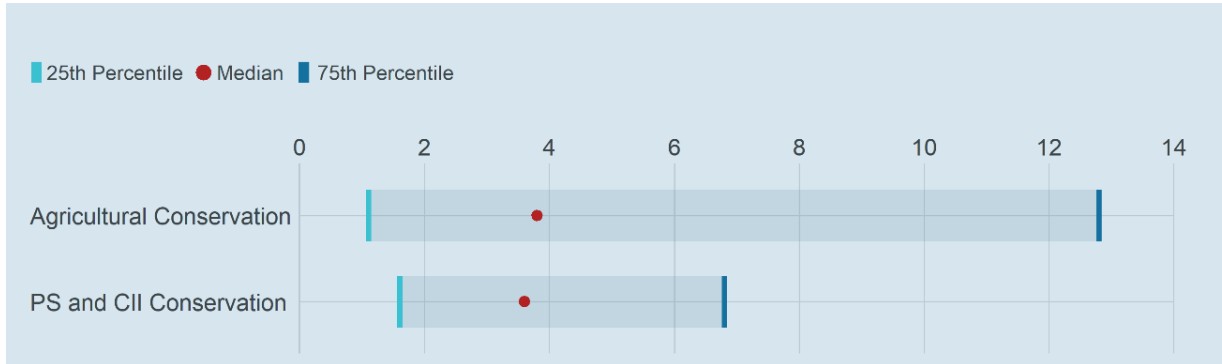

**Figure 6.** Estimated Project Expenditures per Unit of Capacity for Water Conservation Projects (million USD 2021 per MCM). Data source: 2020 Annual Status Report on Regional Water Supply Planning, Florida Department of Environmental Protection (DEP) [6].

At the statewide level, the water supply investment cost change associated with a wider use of water use efficiency measures is even more uncertain than at the regional-level results since the median costs to conserve water might be higher than the costs to expand the water supply in some water supply regions (Figures 4 and 5). In water supply planning regions where (expensive) reclaimed water is the leading water supply option, increasing the use of water conservation practices through water pricing and incentives would probably be more cost-effective rather than solely relying on (expensive) reclaimed water to meet the future water demand. Strategies like water conservation should be used in conjunction with other types of projects to reduce the risk as shown by an application of MPT.

### 3.6. Portfolio Solutions for Developing Water Supply Projects

Figure 7 is an illustrative efficient frontier for computing the shares of each project type (i.e., reclaimed water for potable offset, brackish groundwater, and groundwater recharge) to form the minimum variance portfolio based on a sample of fifty-seven projects (nineteen of each project type). It is ideal to build an efficient frontier using the total net return per unit of capacity for water supply projects. Herein, we use capital costs to make this efficient frontier with the assumption that the benefits per unit capacity are the same for any water supply project. This assumption makes the variance of the capital costs equal to the variance of the returns.

The illustrative simulations show that using MPT instead of simple (equal weight) diversification in investing in new water supply projects can achieve the same level of water supply per dollar spent while considerably reducing the risk (measured as standard deviation of the capital cost). For example, the risk reduces more than three times in SWF region (Table 7). The equal weight means each project type accounts for one-third of the total number of water supply projects that will need to be developed to meet the future water demand. The meaning of the minimum variance is that MPT is used to construct diversified portfolios that minimize the variance/standard deviation of project costs. This finding has important implications for developing new water supply projects to meet future demand. Using MPT, we echo the findings from previous studies such as [19,42,53–56] that show a diversified combination of water supply options likely provides a more reliable water supply and has a lower financial risk of developing it.

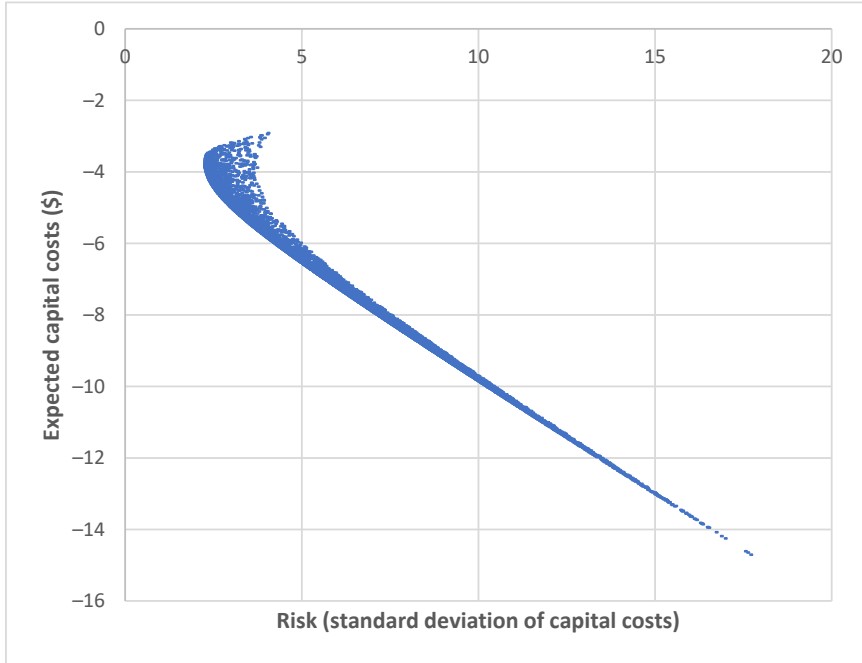

**Figure 7.** An illustrative efficient frontier for computing the weights of each project type to form the minimum variance portfolio based on 10,000 simulations using MPT.

**Table 7.** The standard deviation of capital cost per one unit of capacity for a combination of individual project types.

| Region | Equal Weight | Minimum Variance |
|---|---|---|
| SWF | 6.12 | 1.57 |
| SF | 1.77 | 1.53 |
| Statewide | 5.12 | 3.13 |

It is worth noting that this is an illustrative simulation based on the cost data coming from a sample of water supply projects, and the costs only include capital investments. Thus, to ensure the robustness of the MPT method, one should ideally account for all the costs associated with the projects considered. Additionally, to compute the net return correctly, we need to have an accurate estimation of the total benefits of the projects considered.

## 4. Conclusions

This study provides projections of water demand and supply, and the expenditures needed to meet the future water demand by 2040 in the state of Florida. Using Chebyshev's Inequality, we show that the state likely needs an additional investment of USD 1.11–1.87 billion to meet the future water demand by 2040 because the current water supply capability is unlikely to meet the rising water demand. The lower and upper bounds of the total projected expenditures are equal to around USD 1.646 and USD 3.212 billion, respectively, when accounting for the project expenditures by the projects in design, construction, and on hold (USD 647.18 million). This projected expenditure excludes the expenditure needed to meet the potential increased water allocation to protect and restore the ecosystem. In addition, this expenditure represents the capital costs of a single phase of the water supply projects (i.e., construction) typically funded by the state or water management districts. These costs do not account for the costs of planning design, permitting, transmission, rehabilitation, or replacement of existing facilities and systems (or ongoing operation and maintenance costs). Notwithstanding these limitations, the empirical findings in this study provide a new understanding of the bounds of the investment costs for water supply projects when little information is available to infer the distribution of the costs. The uniqueness of this study compared with the previous study is the use of a probabilistic approach (i.e., Chebyshev's Inequality) to quantify the uncertainty around the costs needed to meet future water demand. The approach is computationally straightforward. It is easy to implement because the solutions can be easily obtained by solving a quadratic equation.

The second major finding is that diversification of water supply options reduces the risk (standard deviation of the capital cost) greatly. We found using MPT in allocating the weight of each project type in developing water supply options reduces the standard deviation of capital costs per one unit of capacity by 74% compared to the equal weight allocation. The present study confirms previous findings and contributes additional evidence that suggests a diversified combination of water supply options greatly enhances the resilience of water supply strategies and has a lower financial risk of developing them. Unlike other classes of optimization techniques such as Robust Optimization and Pareto Frontier, MPT is computationally inexpensive and straightforward. MPT can be easily implemented using widely used Microsoft Excel software. In addition, unlike previous studies such as [17,30,31], this study is likely the first one to apply MPT for a very large geographic region, the statewide level, to quantify the risk associated with diversification in alternative water supply infrastructure investment to meet projected future water demand.

In this study, we present the water demand projections for the baseline scenario, which relies on "status quo" per capita water use, as well as population projections. Additional water conservation programs can reduce future water use; however, the data are not sufficient to estimate the water use reductions beyond the relatively expensive conservation projects discussed in this paper. Further, while WMDs are required to estimate water demand for drought scenarios, corresponding estimates for water supplies during droughts

are not available, limiting the opportunities to explore investments in drought resiliency. Therefore, there is abundant room for further progress in determining all the costs and benefits of water supply projects. As noted earlier, Chebyshev's Inequality tends to provide larger, more conservative, bounds. A future study using other approaches to provide a narrower range of the costs is therefore suggested as time progresses and more data become available. Finding factors driving future water demand and expenditure is not the focus of our study. Thus, further research should be done to forecast the future water demand using different approaches that account for the changing social, economic, and climate conditions to provide a better understanding of the dynamics of the water demand and supply in the state and provide a more accurate expenditure forecast. Herein, we present an estimation of the expenditures associated with an ad hoc policy used to address water shortages through the expansionary water supply phase. The necessity of demand-side solutions to improve water efficiency should be considered in future research.

**Author Contributions:** Conceptualization, D.T. and T.B.; methodology, D.T. and T.B.; software, D.T.; validation, D.T., T.B. and K.B.; formal analysis, D.T.; data curation, D.T.; writing—original draft preparation, D.T.; writing—review and editing, D.T., T.B. and K.B.; visualization, D.T.; supervision, D.T. and T.B. All authors have read and agreed to the published version of the manuscript.

**Funding:** This research received no external funding.

**Data Availability Statement:** All the data used in this study are publicly available from the Florida Water Management Districts, The Florida Department of Agriculture and Consumer Services, and the Florida Department of Environmental Protection.

**Acknowledgments:** This study was supported in part by the U.S. Department of Agriculture, Economic Research Service. Any opinions, findings, conclusions, or recommendations expressed in this publication are those of the authors and should not be construed to represent any official USDA or U.S. Government determination or policy.

**Conflicts of Interest:** The authors declare no conflict of interest. Any opinions, findings, conclusions, or recommendations expressed in this material are those of the author(s) and do not necessarily reflect the views of the Office of Economic and Demographic Research, Florida Legislature, and/or USDA or U.S. Government determination or policy. Additionally, these estimations and forecasts presented in this material should only be considered at the statewide level and are not appropriate for any regional or permitting use.

## Appendix A

*Appendix A.1 Data Used in the Costs Analysis*

The project options identified in the current regional water supply plans (RWSPs), projects being implemented, and projects funded in the past, are summarized in Appendix C of Annual Water Supply Planning published by Florida's s Department of Environmental Protection (FDEP) [6]. Appendix C is publicly available through the FDEP website (https://fdep.maps.arcgis.com/apps/MapSeries/index.html?appid=432a39dd369e4c8 7936fd89bfec40d28; Accessed on 21 May 2022). The project options identified in the current RWSPs projects are being on-hold, implemented (under construction), and projects funded in the past. The project appendix is a spreadsheet, with 1694 rows describing "project items" and 52 columns summarizing various project characteristics. The appendix is the most comprehensive statewide dataset of the Florida water supply and water resource development projects currently available. Appendix C has all information needed for us to compute the costs associated with the first phase (investment phase), including project type, project status, total costs, total capacity, construction beginning date, construction completing date, and location.

To convert the costs to the USD 2021 value, we use Engineering News Record Construction Cost Index History [48]. The index is downloadable at https://www.enr.com/economics/historical_indices/construction_cost_annual_average?_ga=2.167836713.905400 063.1671116882-287204005.1671116882. Accessed on 11 March 2022.

The estimated expenditures for reclaimed water projects account for the beneficial offset being only 0.55 of the actual project capacity [49]. The most current recent reuse water inventory is downloadable at https://floridadep.gov/water/domestic-wastewater/documents/2021-reuse-inventory-all-appendices-excel. Accessed on 21 May 2022

*Appendix A.2 Methods Used in Water Demand Projections*

Under Section 373.036, Florida Statutes, the governing board of each WMD must develop a district water management plan. If it is determined that there will be water shortages during the period 2020–2040, WMDs are required to develop RWSPs. Each RWSP contains water supply development project options and water resource development projects and programs. In these RWSPs, the methods used in estimating and forecasting water demand are fully described. All the RWSPs are downloadable from the five Florida water management district's websites. Herein, we provide a summary of the methods that WMDs used to estimate and forecast water use/demand.

Appendix A.2.1 Public Supply (PS) Water Use

Florida's Water Management Districts (WMDs) use a similar approach to project future water demand. The estimated base year water use is typically equivalent to a utility's reported pumpage. WMDs estimate water use for the suppliers' service area with an allocation above 0.1 million gallons per day (mgd). Public supply (PS) is projected by relying on the "unit water demand" approach. The PS water demand is equal to a "unit water demand coefficient" (e.g., water use per capita) multiplied by the number of users [6,11–13,57–60]. The unit water demand is typically computed as the water use per capita per year or average the rate over a number of years. The approach allows WMDs quickly update the water use projections with newly available data. The number of users refers to the population served within the service areas of the suppliers. All of the WMDs utilize the Bureau of Economic and Business Research (BEBR) county population projections in developing the PS forecasts. The main difference regarding the method used to project future PS water use is the population projections. The publication years for the population estimates utilized by the WMDs range from 2015 to 2020 (with the base population year being 2014 through 2019) [6,11–13,57–60].

Appendix A.2.2 Domestic Self-Supply (DSS) Water Use

WMDs estimate and project DSS water use for (a) small public supply systems (i.e., those smaller than 0.1 mgd in the permitted capacity or pumpage), and (b) residential dwellings systems that are provided water from a dedicated, on-site well and are not connected to a central utility. For small public supply systems, the method used to estimate and project DSS water use is the same as methods used for PS water use' estimations and projections. WMDs also use similar methods (i.e., unit water demand) to estimate and project the per capita water use multiplied by the estimated population. The PS per capita rate includes all types of uses served by the public supply, including household use, commercial use, and others. Many of the uses are not relevant to DSS, and therefore, the residential per-capita rate is estimated for the PS sector and then it is applied to DSS. Residential per-capita also referred to as household water use rate, is generally based on the residential water use allocation from relevant consumptive use permits (CUPs) or water use permits [6,11–13,57–60].

Appendix A.2.3 Landscape/Recreational (L/R) Water Use

The Landscape/Recreational (L/R) category includes such users as self-supplied golf courses, parks (including water parks), and commercial center irrigation. North West Florida Water Management District (NWFWMD) also includes residential irrigation wells in this water use category. The methods used to estimate and project water use for this category are also similar to the methods used for PS water use projections in a sense that base year water use and population projections from BEBR. Specifically, the base year water

use typically refers to the water use estimated from reported and audited pumpage. The changes in water use for the golf course irrigation over time are based on the growth rate (either as suggested by the industry and local planning councils or as estimated using a golf course irrigation model). To project water demand for non-golf demand, it is typically assumed to grow at the rate of increase for the BEBR-medium population. If the residential irrigation wells are accounted for in this water use category, the estimated water use per day is equal to the number of well time 76 gallons per day (1 gallon = 3.785 L).

Appendix A.2.4 Commercial/Industrial/Institutional (CII) Water Use

The category includes all reporting commercial, industrial, and institutional (CII) self-supplied permittees (including mining and dewatering uses). Only consumptive uses are included (i.e., recycled surface water and non-consumptive uses excluded). WMDs typically use reported pumpage to estimate base-year water use. WMDs use three different methods to project water use for this category. The first method is a survey-based method. That is, WMDs request projections from the permittees directly. NWFWMD largely relies on this method to project the CII water use.

The second method is used by the SWFWMD (excluding the portion in the CFWI). The district determined that water use is generally correlated with the county's one-year cross-regional product (GRP) growth rate from Woods and Poole (2017). The only exception is Mosaic water use, for which the company provided growth projections for its processing facilities and mining operations.

The third method is used by The SFWMD, SJRWMD, and SRWMD. The method comes from the assumption that CII water use follows population trends. For example, in the CFWI, NFRWSP, and SRWMD, the county-specific five-year average gallon per capita per day is based on the USGS data or the calibration dataset from the East-Central Florida Transient (ECFTX) groundwater model. This per capita rate is then multiplied by the BEBR-medium population projection growth rate.

Appendix A.2.5 Power Generation (PG) Water Use

In all the WMDs, this category includes water used for power generation facilities not supplied by the PS (primarily, thermoelectric power). For thermoelectric power generation, net water use for thermoelectric power generation may include on-site potable uses, as well as water loss due to evaporation, blowdown, drift, and leakages. The consumptive water use in the sector is small because most of the water used for recirculation and cooling is returned to the same water body. Therefore, these projections unlikely significantly alter the overall statewide water demand projections.

WMDs often use three methods to estimate and project PG water use. The PG water use forecast in SFWMD is established largely relying on information from the power generation facilities owners and managers (such as Florida Power and Light). NWFWMD water use projections for this category are largely based on the reports from permittees. In SRWMD and NFRWSP, the forecasts are based on the ten-year site plans and the BEBR population projections. SWFWMD and CFWI rely on 10-year site plans and electricity demand projections to project future water demand for this category. These two WMDs use the average water use per megawatt and then use population projections to project future PG water needs.

Appendix A.2.6 Water Demand for Agriculture (AG)

Florida's Department of Agricultural and Consumer Services (FDACS) establish an agricultural water supply planning program that includes "the development of data indicative of future agricultural water supply demands," based on at least a 20-year planning period. FDACS's Florida Statewide Agricultural Irrigation Demand (FSAID) geodatabase provides the agricultural acreage and water use projections for each WMD and planning region. This information is updated annually and is publicly available [45]. The method and data used to project the water used are explained at length and are accessible to the

general public at the FDACS website [45] (https://www.fdacs.gov/Agriculture-Industry/Water/Agricultural-Water-Supply-Planning. Accessed on 25 June 2022). In general, the water projections are split into four categories: irrigation, livestock watering, frost-freeze protection, and aquaculture.

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
