# Peer review of "The Cost of Alternative Water Supply and Efficiency Options under Uncertainty: An Application of Modern Portfolio Theory and Chebyshev’s Inequality"

_2673-4834, doi:10.3390/earth4010003_

Round 1
Reviewer 1 Report (New Reviewer)
This study reports on the cost of alternative water supply and efficiency options under uncertainty. The analysis of the article sounds reasonable. My (minor) concern is related to the novelty introduced by the authors, which should be stressed further (especially in the concluding remarks) before the next submission. In addition, authors are advised to move the appendix into the main body of the manuscript.
Author Response
We appreciate the time and effort that you dedicated to providing feedback on our manuscript. We are grateful for your comments as they lead to valuable improvements to our paper. Our responses to the suggestions appear in italics below. Attached is the revised manuscript with all changes/revisions tracked.
This study reports on the cost of alternative water supply and efficiency options under uncertainty. The analysis of the article sounds reasonable. My (minor) concern is related to the novelty introduced by the authors, which should be stressed further (especially in the concluding remarks) before the next submission. In addition, authors are advised to move the appendix into the main body of the manuscript.
Response: This is a great suggestion, thank you! We have moved the appendix into the main boby. We have also added additional information about the methods used to project future water use. In addition, we have added and revised text into the introduction and conclusion sections to further highlight the main novelty of this study.
The revised text to the Introduction reads as: “Despite an extensive literature on the use of MPT in natural resources and environmental management field, it remains unclear, though, the extent to which uncertainty of the costs of alternative water supply affect the total investment costs needed to meet the future water demand, and how water utility could reduce the financial risks associated with the uncertainty. In this paper, we perform the first large-scale analysis to compare different water supply options to improve water security and address water supply cost uncertainty in the context of Florida’s natural resources, laws, and regulations. Much like the approach used in other U.S. states such as California and Texas [34, 35], Florida’s water agencies tend to rely on a scenario-based water supply investment planning, in which specific construction costs and capacities are assumed for future water supply projects. To enhance the scenario-based planning approach, this paper proposes to use the Modern Portfolio Theory (MPT) [36], which allows the selection of diversified water supply options to reduce the impact of capital cost uncertainty and manage the risk. We illustrate how using MPT can reduce the overall risk of the total portfolio on investments in developing alternative water supply options to meet future water use. The approach relies on a principle that maximizes the expected investment returns for a given level of uncertainty (e.g., variance or standard deviation of the investment returns) or minimizes uncertainty for a given expected level of return. In addition, to account for the uncertainty of the projected expenditures, we use Chebyshev’s Inequality, which allows estimating probability bounds based only on the mean and variance of an unidentified distribution [37, 38]. We then apply this framework to estimate the range of expenditures needed to meet projected future water demand for each of Florida’s regions by 2040.”
The second revised text to the Introduction reads as: “The first major contribution of this study, despite its somewhat exploratory nature, this study offers insights into the extent to which diversification of water supply options reduces financial risk. This study contributes to the existing literature on MPT application to the choice of water supply investment strategies. Specifically, unlike previous studies (and particularly, [39]), this analysis examines the uncertainty associated with capital costs of alternative water supply investment options. The study has confirmed the findings of [16, 31, 32], which found that diversification of water supply options increases the resilience of water supply systems by reducing their financial risks. The second contribution of this study is that this study proposes a new method of accounting for cost uncertainty that can inform the process of policy and financial planning. While it is ideal to know the distribution of the costs when inferring the upper and lower bounds (e.g., 95% interval) of the costs, such analysis typically requires an extensive dataset to identify the type of the distribution, which is rarely available to the researchers or water supply planners. A key strength of this study is the use of Chebyshev’s Inequality to account for the capital costs uncertainty. The use of Chebyshev’s Inequality does not require an assumption about the type of distribution of the costs that might follow [37, 38]. To the best of our knowledge, this study has demonstrated, for the first time, that one can use Chebyshev’s Inequality to quantify the uncertainty of the investments needed to meet rising water demand with limited data. The uniqueness of this study compared with the previous studies is the introduction of a probabilistic-based approach to quantify the uncertainty of the investment costs to meet future water demand. Previous studies often provide estimations of the costs using descriptive statistical techniques [40, 41], optimization-based [14, 15, 17, 22, 28], or econometric approach [39], and overlook the uncertainty of the costs and the extent to which the uncertainty affect the estimations [15, 42].”
The revised texts to the Conclusion section reads as: “The first major contribution of this study, despite its somewhat exploratory nature, this study offers insights into the extent to which diversification of water supply options reduces financial risk. This study contributes to the existing literature on MPT application to the choice of water supply investment strategies. Specifically, unlike previous studies (and particularly, [39]), this analysis examines the uncertainty associated with capital costs of alternative water supply investment options. The study has confirmed the findings of [16, 31, 32], which found that diversification of water supply options increases the resilience of water supply systems by reducing their financial risks. The second contribution of this study is that this study proposes a new method of accounting for cost uncertainty that can inform the process of policy and financial planning. While it is ideal to know the distribution of the costs when inferring the upper and lower bounds (e.g., 95% interval) of the costs, such analysis typically requires an extensive dataset to identify the type of the distribution, which is rarely available to the researchers or water supply planners. A key strength of this study is the use of Chebyshev’s Inequality to account for the capital costs uncertainty. The use of Chebyshev’s Inequality does not require an assumption about the type of distribution of the costs that might follow [37, 38]. To the best of our knowledge, this study has demonstrated, for the first time, that one can use Chebyshev’s Inequality to quantify the uncertainty of the investments needed to meet rising water demand with limited data. The uniqueness of this study compared with the previous studies is the introduction of a probabilistic-based approach to quantify the uncertainty of the investment costs to meet future water demand. Previous studies often provide estimations of the costs using either descriptive statistical techniques [40, 41], optimization-based [14, 15, 17, 22, 28], or econometric approach [39], and overlook the uncertainty of the costs and the extent to which the uncertainty affect the estimations [15, 42].”
In the introduction, we have also added an additional paragraph reviewing the methods used to address uncertainty in water resources planning. The paragraph reads as “Planning for sustainability is a complex task because of uncertainties around the water demand, water supply, and investment costs needed to balance water demand and supply (e.g., costs to increase water supply or costs to reduce water demand per capita through water conservation). Fund and water managers both face a similar challenge. They both need to have reliable water systems to meet water demand while sources of investment and water vary randomly [14-19]. Several approaches have been used to deal with the water supply and demand uncertainties: scenario-based robust optimization (RO) [14, 20, 21], adaptive pathways (AP) [22-24], and real options analysis (ROA) [15, 25, 26]. In general, these approaches are ruled-based planning frameworks [15, 22]. Thus, to take uncertain future conditions into account, researchers need to somehow quantify the uncertainties (e.g., through a probability distribution and an ensemble of realizations [26, 27]) and then embed these probability distributions or realizations into the model. Thus, ROA is considered to be impractical without pre-defined distributions or realizations [15, 27, 28]. The second challenge of using ROA is that the method is not well-suited for quantifying the trade-off between the return and risks associated with various portfolio compositions [17, 27, 29].”

Reviewer 2 Report (New Reviewer)
This manuscript has a good novelty, clear objectives and questions. Very well written and very valuable results. However, paying attention to the following points will improve it.
1- Novelty should be clearly stated in the introduction.
2- The literature review is a little weak. Please improve.
3- The results should be presented visually. Simply presenting the results in text form is not enough. Presenting graphs and images will improve the understanding of the results.
4- The data used should be presented in detail. It is very general.
5- The methodology of forecasting is presented in summary. It should be presented in detail.
Author Response
Thank you so much for the helpful suggestions. These have improved the manuscript. Our responses appear in italics. Attached is the revised manuscript with all changes/revisions tracked.
This manuscript has a good novelty, clear objectives and questions. Very well written and very valuable results. However, paying attention to the following points will improve it.
1.Novelty should be clearly stated in the introduction.
Response: We think that this is an excellent suggestion. We have revised the introduction and also added additional text into the introduction and conclusion sections to further highlight the novelty of this study.
The revised text to the Introduction reads as: “ Despite an extensive literature on the use of MPT in natural resources and environmental management field, it remains unclear, though, the extent to which uncertainty of the costs of alternative water supply affect the total investment costs needed to meet the future water demand, and how water utility could reduce the financial risks associated with the uncertainty. In this paper, we perform the first large-scale analysis to compare different water supply options to improve water security and address water supply cost uncertainty in the context of Florida’s natural resources, laws, and regulations. Much like the approach used in other U.S. states such as California and Texas [34, 35], Florida’s water agencies tend to rely on a scenario-based water supply investment planning, in which specific construction costs and capacities are assumed for future water supply projects. To enhance the scenario-based planning approach, this paper proposes to use the Modern Portfolio Theory (MPT) [36], which allows the selection of diversified water supply options to reduce the impact of capital cost uncertainty and manage the risk. We illustrate how using MPT can reduce the overall risk of the total portfolio on investments in developing alternative water supply options to meet future water use. The approach relies on a principle that maximizes the expected investment returns for a given level of uncertainty (e.g., variance or standard deviation of the investment returns) or minimizes uncertainty for a given expected level of return. In addition, to account for the uncertainty of the projected expenditures, we use Chebyshev’s Inequality, which allows estimating probability bounds based only on the mean and variance of an unidentified distribution [37, 38]. We then apply this framework to estimate the range of expenditures needed to meet projected future water demand for each of Florida’s regions by 2040.”
The second revised text to the Introduction reads as: “The first major contribution of this study, despite its somewhat exploratory nature, this study offers insights into the extent to which diversification of water supply options reduces financial risk. This study contributes to the existing literature on MPT application to the choice of water supply investment strategies. Specifically, unlike previous studies (and particularly, [39]), this analysis examines the uncertainty associated with capital costs of alternative water supply investment options. The study has confirmed the findings of [16, 31, 32], which found that diversification of water supply options increases the resilience of water supply systems by reducing their financial risks. The second contribution of this study is that this study proposes a new method of accounting for cost uncertainty that can inform the process of policy and financial planning. While it is ideal to know the distribution of the costs when inferring the upper and lower bounds (e.g., 95% interval) of the costs, such analysis typically requires an extensive dataset to identify the type of the distribution, which is rarely available to the researchers or water supply planners. A key strength of this study is the use of Chebyshev’s Inequality to account for the capital costs uncertainty. The use of Chebyshev’s Inequality does not require an assumption about the type of distribution of the costs that might follow [37, 38]. To the best of our knowledge, this study has demonstrated, for the first time, that one can use Chebyshev’s Inequality to quantify the uncertainty of the investments needed to meet rising water demand with limited data. The uniqueness of this study compared with the previous studies is the introduction of a probabilistic-based approach to quantify the uncertainty of the investment costs to meet future water demand. Previous studies often provide estimations of the costs using descriptive statistical techniques [40, 41], optimization-based [14, 15, 17, 22, 28], or econometric approach [39], and overlook the uncertainty of the costs and the extent to which the uncertainty affect the estimations [15, 42].”
The revised text to the Conclusion section reads as: “This study provides projections of water demand and supply, and the expenditures needed to meet the future water demand by 2040 in the state of Florida. Using Chebyshev’s Inequality, we show that the state likely needs an additional investment of $0.999-$2.565 billion to meet the future water demand by 2040 because the current water supply capability is unlikely to meet the rising water demand. The lower and upper bounds of the total projected expenditures are equal to around $1.646 and $3.212 billion, respectively, when accounting for the project expenditures by the projects in design, construction, and on hold ($647.18 million). This projected expenditure excludes the expenditure needed to meet the potential increased water allocation to protect and restore the ecosystem. In addition, this expenditure represents the capital costs of a single phase of the water supply projects (i.e., construction) typically funded by the state or water management districts. These costs do not account for the costs of planning design, permitting, transmission, rehabilitation, or replacement of existing facilities and systems (or ongoing operation and maintenance costs). Notwithstanding these limitations, the empirical findings in this study provide a new understanding of the bounds of the investment costs for water supply projects when little information is available to infer the distribution of the costs. The uniqueness of this study compared with the previous study is the use of a probabilistic approach (i.e., Chebyshev’s Inequality) to quantify the uncertainty around the costs needed to meet future water demand. The approach is computationally straightforward. It is easy to implement because the solutions can be easily obtained by solving a quadratic equation.
The second major finding is that diversification of water supply options reduces the risk (standard deviation of the capital cost) greatly. We found using MPT in allocating the weight of each project type in developing water supply options reduces the standard deviation of capital costs per one unit of capacity by 74% compared to the equal weight allocation. The present study confirms previous findings and contributes additional evidence that suggests a diversified combination of water supply options greatly enhances the resilience of water supply strategies and has a lower financial risk of developing them. Unlike other classes of optimization techniques such as Robust Optimization and Pareto Frontier, MPT is computationally inexpensive and straightforward. MPT can be easily implemented using widely used Microsoft Excel software. In addition, unlike previous studies such as [17, 30, 31], this study is likely the first one to apply MPT for a very large geographic region, the statewide level, to quantify the risk associated with diversification in alternative water supply infrastructure investment to meet projected future water demand.”
2. The literature review is a little weak. Please improve.
Response: We agree with this assessment. We have added a paragraph to the literature review section. The paragraph summarizes the methods commonly used to address uncertainty in water resources planning. The paragraphs read as “Planning for sustainability is a complex task because of uncertainties around the water demand, water supply, and investment costs needed to balance water demand and supply (e.g., costs to increase water supply or costs to reduce water demand per capita through water conservation). Fund and water managers both face a similar challenge. They both need to have reliable water systems to meet water demand while sources of investment and water vary randomly [14-19]. Several approaches have been used to deal with the water supply and demand uncertainties: scenario-based robust optimization (RO) [14, 20, 21], adaptive pathways (AP) [22-24], and real options analysis (ROA) [15, 25, 26]. In general, these approaches are ruled-based planning frameworks [15, 22]. Thus, to take uncertain future conditions into account, researchers need to somehow quantify the uncertainties (e.g., through a probability distribution and an ensemble of realizations [26, 27]) and then embed these probability distributions or realizations into the model. Thus, ROA is considered to be impractical without pre-defined distributions or realizations [15, 27, 28]. The second challenge of using ROA is that the method is not well-suited for quantifying the trade-off between the return and risks associated with various portfolio compositions [17, 27, 29].”
3. The results should be presented visually. Simply presenting the results in text form is not enough. Presenting graphs and images will improve the understanding of the results.
Response: Thank you for pointing this out. We have converted table 3 into a figure. We acknowledge that converting other tables in figures would greatly improve the readability of the paper. However, tables with numbers in different units (e.g., cubic meters for water demand and supply, and US dollars for costs) are challenging to be converted to figures. Also, some regions have very low (even zeros) water shortages and/or expenditures, while other regions with high water shortages and/or expenditures. The differences make it harder to present all these information in a figure. We opt to keep these tables as they are.
4. The data used should be presented in detail. It is very general.
Response: Thank you for pointing this out. We have added additional descriptions of the data into the Appendix. The descriptions are divided into two main sections: data used in the costs/expenditures analysis and methods used in water demand projections.
5. The methodology of forecasting is presented in summary. It should be presented in detail.
Response: We have added an additional description of the methods used to project water demand in the Appendix.

Reviewer 3 Report (New Reviewer)
The Cost of Alternative Water Supply and Efficiency Options Under Uncertainty: An Application of Modern Portfolio Theory and Chebyshev’s Inequality
The article has, rather, an economic nuance than a technical, hydrological, hydro-edilitar and scientific one.
The authors prove to be well competent in regional and zonal issues/studies related to supply water use and it efficiency.
The applicative character of the article is much higher than the scientific one, being of real use to regional and zonal authorities.
Author Response
Thank you so much for the suggestions you provided for our manuscript. We have revised the manuscript accordingly, and we appreciate you helping us improve the manuscript. We describe revisions made to address the comments below, in italics. Attached is the revised manuscript with all changes/revisions tracked.
The Cost of Alternative Water Supply and Efficiency Options Under Uncertainty: An Application of Modern Portfolio Theory and Chebyshev’s Inequality
The article has, rather, an economic nuance than a technical, hydrological, hydro-edilitar and scientific one. The authors prove to be well competent in regional and zonal issues/studies related to supply water use and it efficiency. The applicative character of the article is much higher than the scientific one, being of real use to regional and zonal authorities.
Response: Thank you for the feedback. We have revised the manuscript considerably. First, we have moved the appendix into the main text. Second, we have also added additional information about the methods used to project future water demand and the data sources for the costs analysis. Third, we have added additional texts into the introduction and conclusion sections to further highlight the novelty of this study.

This manuscript is a resubmission of an earlier submission. The following is a list of the peer review reports and author responses from that submission.
Round 1
Reviewer 1 Report
The present manuscript focuses on important Topis for the sustainability of drinking water systems (investment planning and water conservation). Although, several improvements are recommended for a sound scientific paper, namely:
- the introduction should be improved with more scientific references, namely about methods for uncertainty assessment. Most of the references refer to reports about Florida.
- Moreover, the study lacks information about the aspects that led to the adoption of methods used in the paper.
- improve description methods description, namely about the Modern Portfolio Theory and the portfolio adopted.
- Discussion lacks reference for previous studies for cross-comparison.
- The discussion of the figures should be improved for more understanding.
- several concepts lack consistency, e.g., water need, demand, supply, and shortage. Therefore, the most relevant concepts should be presented in the introduction e used carefully throughout the paper.
- For example, Figure 2 scheme interpretation is not clear. In 2020, when supply is higher than demand it is possible to have a shortage?
- The paper should also make clearer the main contributions of the research. Otherwise, it corresponds only to a technical report.
Reviewer 2 Report
From my perspective as an engineer, applying Modern Portfolio Theory to reduce the impact of capital cost uncertainty and manage the risk is very interesting. Some details to improve are:
a. It would be worthwhile to contrast the water use data for the agricultural sector with another source, for example, the Florida Department of Agriculture and Consumer Services (FDACS) annual estimated water use update report, available here:
https://www.fdacs.gov/content/download/84471/file/FSAID-VI-Water-Use-Estimates-Final-Report.pdf.
Since their data are different and highlight the importance that despite being different, they can illustrate how water consumption for agricultural purposes is lower than for human consumption. This fact gives more weight to the hypothesis that Florida, concerning the general trend of water uses worldwide, is an outlier since it is known that agricultural uses correspond to most usage.
b. The data sources used to create Figure 3 and Tables 3 and 4 are not directly mentioned. I could not find how these evolutions of Water Demands could be affected (decreased) by the Estimated Efficiency Improvements by the Water Management District from the point of view of agriculture; and by the incremental improvement of the efficiency of the potable water distribution systems, as well as the reduction of water consumption by the citizenry. Residential water use per person in Florida has declined since 2000 due to water conservation efforts, water restrictions, the increased use of reclaimed water, and "Florida-Friendly" landscaping techniques.
c. There are many misspellings in the subchapter headings. For example: "3.5.PortfolioSolutionsfordevelopingalternativewatersupply. projects ".
d. It is unclear how the results are obtained for the Statewide scope in Table 8. Is it an average? Does it make statistical sense to have a standard deviation related by region?
e. Please clarify the sources for Figures 4 and 5; in the images, it says DEP (2021a). Is the source the Florida Department of Environmental Protection? Since it is the source of many assumptions on which the research development is based, they must be modified and adapted to the document's format; and specify exactly their source. If it is an unpublished report, the report cannot use this or these kinds of references.
f. The Materials and Methods chapter does not sufficiently explain how to replicate and obtain the results of Tables 3, 4, 5, 6 and 7. An excellent example of how this should be presented in the case of the variance of the portfolio is illustrated in Appendix 1.
Reviewer 3 Report
Dear Authors,
Thank you for your interesting article. Enclosed are some comments:
- in line 182 the abbreviation RWSP is used, which is not explained in the text
- in the title for table 1, the word "investigated" could be inserted at the beginning
- please check the layout of table 2 (pagination and alignment)
- in line 402 please check grammar
- check heading for chapter 3.5
- You discuss diversification of water supply projects as a way to reduce cost risk. Deciding on a particular water supply option usually also depends on certain local and technical conditions. Please comment on how you rate the importance of the technical and economic aspects.